



# Algorithm to retrieve aerosol optical properties using lidar measurements on board the EarthCARE satellite

Tomoaki Nishizawa[1], Rei Kudo[2], Eiji Oikawa[2], Akiko Higurashi[1], Yoshitaka Jin[1], Nobuo Sugimoto[1], Kaori Sato[3], Hajime Okamoto[3]

[1]National Institute for Environmental Studies, 16-2 Onogawa, Tsukuba, 305-8506, Japan
[2]Meteorological Research Institute, Japan Meteorological Agency, 1-1 Nagamine, Tsukuba, 305-0052, Japan
[3]Research institute for applied mechanics, Kyushu university, 6-1 Kasuga-koen, Fukuoka, 816-8580, Japan

*Correspondence to*: Tomoaki Nishizawa (nisizawa@nies.go.jp)

**Abstract.**

Algorithms were developed to produce ATLID (Atmospheric Lidar) L2 aerosol products using ATLID L1 data. The algorithms estimated the following four products: (1) Layer identifiers such as aerosols, clouds, clear-skies, or surfaces (feature masks) were estimated by the combined use of vertically variable criteria and spatial continuity methods developed for the CALIOP (Cloud-Aerosol Lidar and Infrared Pathfinder Satellite Observation) analysis. (2) Aerosol optical properties such as extinction coefficient, backscatter coefficient, depolarization ratio, and lidar ratio at 355 nm were estimated by our developed optimization method using the Gauss-Newton method combined with the line search method developed for ground-based measurements. (3) Six aerosol types, namely smoke, pollution, marine, pristine, dusty-mixture, and dust were identified based on a two-dimensional diagram of the lidar ratio and depolarization ratio at 355 nm developed by cluster-analysis using the AERONET (AErosol RObotic NETwork) dataset with ground-based lidar data. (4) The planetary boundary layer height was determined using the improved wavelet covariance transform method for the ATLID analysis. We evaluated the algorithm's performance using simulated ATLID L1 data generated by Joint-Simulator (Joint Simulator for Satellite Sensors), incorporating aerosol and cloud distributions from numerical models. Results from applying the algorithms to the simulated ATLID L1 data with realistic signal noise added for aerosol or cloud predominant cases revealed: (1) misidentification of aerosol and cloud layers by the feature mask algorithm was relatively low, approximately 10%; (2) the retrieval errors of aerosol optical properties were $0.08\times10^{-7}\pm1.12\times10^{-7}\,\mathrm{m^{-1}sr^{-1}}$ ($2\pm34\%$ in relative error) for backscatter coefficient and $0.01\pm0.07$ ($4\pm27\%$) for depolarization ratio; (3) aerosol type classification was generally performed well, with a 37% of misclassification for dust. These results indicate that the algorithm's capability to provide valuable insights into the global distribution of aerosols and clouds, facilitating assessments of their climate impact through atmospheric radiation processes.

## 1 Introduction

Global measurements of the optical properties of atmospheric particles , encompassing aerosols and clouds, play a pivotal role in evaluating their climatic effects through atmospheric radiation processes. For instance, the presence of highly light-





absorbing aerosols such as black carbon above or below the cloud layer can prodoundly the radiative transfer process of the upper atmosphere (Takemura et al., 2002; Oikawa et al. 2013, 2018). The Cloud-Aerosol Lidar and Infrared Pathfinder Satellite Observation (CALIPSO) satellite, equipped with the two-wavelength (1064, 532nm) polarization Mie-scattering lidar

CALIOP (Cloud-Aerosol LIdar with Orthogonal Polarization), has globally measured aerosols and cloud distribution over the long term from 2006 to 2023 (Winker et al. 2010). This role will transition to ATLID (Atmospheric Lidar) (Heiliere et al. 2017; do Carmo et al. 2021), the 355nm high-spectral resolution lidar (HSRL) with polarization measurement function onboard the EarthCARE satellite (Earth Cloud Aerosol and Radiation Explorer). EarthCARE is a joint Japan-Europe satellite observation mission by JAXA (Japan Aerospace Exploration Agency), NICT (National Institute of Information and

Communications Technology), and ESA (European Space Agency), which aims to conduct comprehensive global observations of clouds, aerosols, and atmospheric radiation (Wehr et al. 2023). The EarthCARE satellite features a 94 GHz cloud radar (CPR) with doppler measurement capability (Nakatsuka et al. 2023), a multiwavelength imager (MSI), comprising seven channels in the visible to infrared wavelength range (Wallace et al. 2016), and a broadband radiometer (BBR) designed for measuring shortwave and longwave radiation (Wallace et al. 2016). The scientific objectives emcompass (1) observing the

global vertical distributions of natural and anthropogenic aerosols and their interaction with clouds; (2) observing global cloud distributions, cloud-precipitation interactions, and vertical motion characteristics within clouds; (3) evaluating the vertical profiles of radiative heating and cooling of the atmosphere (Illingworth et al. 2015; Okamoto et al. 2024).

ATLID, a HSRL, can independently measure backscattered light from atmospheric particles (Mie signal) and backscattered light from atmospheric molecules (Rayleigh signal) separately, distinguishing it from CALIOP, a Mie-scattering lidar. This

unique feature allows for the independent extraction of the vertical distributions of the extinction coefficient and backscattering coefficients of atmospheric particles by analyzing these two signals. Moreover, ATLID conducts polarization measurements, enabling the extraction of the copolar component, parallel to the laser polarization (copolar component), the cross-polar component (perpendicular to the laser polarization) of the Mie signal at a wavelength of 355 nm, and the Rayleigh signal. Signal calibrations are performed on these measured signals, resulting in  published attenuated backscatter coefficients.

Multichannel lidar measurements, including those of ATLID and CALIOP, facilitate simultaneous understanding of various optical and microphysical properties of atmospheric particles. This includes the identification of major atmospheric layers such as aerosol or cloud layers (Okamoto et al. 2008; Vaughan et al. 2009; Liu et al. 2009; Hagihara et al. 2010) and the identification of aerosol and cloud types such as marine, dust, warm water, or 2-D ice (Omar et al. 2009; Illingworth et al. 2015; Okamoto et al. 2019). Additinoally, the size distribution and refractive index of total aerosols can be estimated using extinction

coefficients and backscatter coefficients at two wavelengths (355 nm and 532 nm) observed by the Raman lidar and HSRL (e.g., Müller et al. 2014). Furthermore, the extinction coefficients of some major aerosol components (such as mineral dust and black carbon) have been estimated (Nishizawa et al. 2017; Kudo et al. 2023).Understanding of the climate impact of aerosols, necessitates not only comprehending the optical and microphysical properties of total aerosols but also those of individual aerosol species. For example, aerosols with strong light absorption properties (e.g., black carbon) have been reported



to influence cloud formation and global atmospheric and water circulations in large fields (Menon et al. 2002; Koren et al. 2004).

The EarthCARE mission generates Level 1 (L1) products, which are physical quantities derived after calibrating the measurements of each instrument, and Level 2 (L2) products, consisting of geophysical variables related to clouds, aerosols, etc, utilising L1 data either independenally or in combination (Eisinger et al. 2024). Each sensor development organization

(JAXA and NICT for CPR, and ESA for others) generates the L1 product, and the L2 product is generated by individual agencies. For example, L2 products using ATLID, such as layer identifiers, particle (aerosol, cloud) type identifiers, and optical properties, such as extinction coefficient, backscatter coefficient, and depolarization ratio, and cloud top height, are estimated from the ESA (Irbah et al. 2023; Donovan et al. 2023a,b; Zadelhoff et al. 2023; Wandinger et al. 2023). Synergy products are also generated by combining L1 data from the other instruments. JAXA also provides various ATLID standalone and

synergistic products, while similar to ESA products, are produced using different independent retrieval methods developed independently.

Based on the above background, we have developed atmospheric particle retrieval algorithms to generate L2 products of JAXA using ATLID L1 data. In this study, we focus on the retrieval algorithms and products related to aerosols, whereas the description of cloud estimation is planned for separate papers. Section 2 describes the products retrieved using the algorithms

and their flows. Section 3 describes the algorithms developed to retrieve the individual products. Section 4 presents the products estimated using the simulated input data (ATLID L1 data) and the performance of the algorithms. In Section 5, we summarize and discuss the prospects.

## 2 Algorithm flow and products

Figure 1 illustrates the products and flow of the algorithms. Initially, the algorithm aims to enhance the signal quality of

ATLID L1 data and reduce the signal noise using a discrete wavelet transform (DWT) [Fang and Huang, 2012]. In this study, two types of Daubechies wavelets (support numbers 2 (D2) and 4 (D4)) are employed. Noise reduction is performed iteratively with D2, followed by D4, and then the noise reduction using D2 is applied again, followed by D4 and so on (D2=>D4=>D2=>D4=>D2...). A simulation analysis with added random noise indicated that the applying the DWT method could improve the signal-to-noise ratio (SNR) by a factor of two or more. Therefore, this method was adopted in the present

study.

To further enhance the SNR, we integrate the data horizontally after applying the DWT method to the minimum horizontal resolution (~0.3 km) data from ATLID L1. This integration yields data at a 1 km horizontal resolution (ATLID L1(1) in Figure 1). Additionally, a moving average of this 1 km horizontal resolution data with a width of 10 km was produced (ATLID L1 (1*) in Figure 1). Consequently, the algorithm uses the three L1 data created (ATLID L1 (0.3), (1), and (1*) in Figure 1), with

an altitude resolution of 0.1 km, akin to the original data, as input data when estimating each product. A quality check (QC) involving the SNR is performed on each of the generated data.





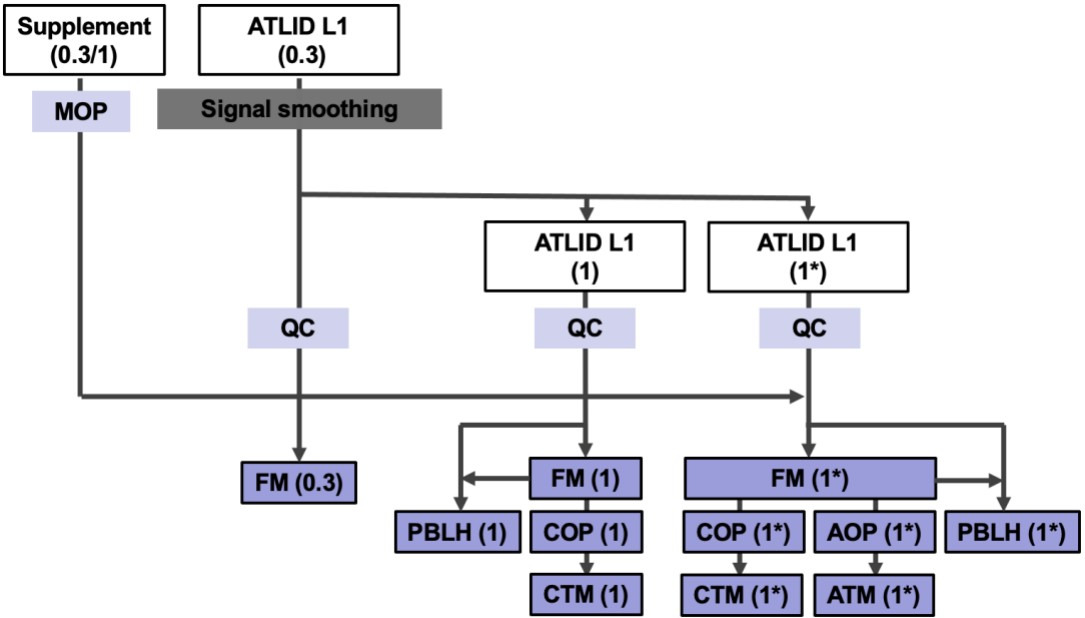

Figure 1. ATLID L2 products and the flow of algorithms.

Six different products (FM (Feature Mask), AOP (Aerosol Optical Properties), COP (Cloud Optical Properties), ATM (Aerosol Target Mask), CTM (Cloud Target Mask), PBLH (Planetary Boundary Layer Height) in Fig. 1) are created from the ATLID L1 data. First, the algorithm identifies which component of atmospheric molecules, aerosols, or cloud particles is predominantly present at each altitude, and outputs its identifier (FM). Additionally, the surface, subsurface, and fully attenuated layers are identified here. Secondly, the particle optical properties (POP), that is, the extinction coefficient ($\alpha_p$),

backscatter coefficient ($\beta_p$), depolarization ratio ($\delta_p$), and lidar ratio ($S_p$) at 355nm, are retrieved from the ATLID L1 data. Whether the derived POP is of aerosol or cloud origin is determined according to the FM product (AOP or COP). Aerosol (ATM) and cloud (CTM) particle types are also identified at each altitude. The height of the planetary boundary layer (PBLH) is then estimated. The product derived from original 0.3 km data is only the FM because of the low SNR. To account for cloud heterogeneity and the low SNR of the aerosol layers, the cloud products (COP and CTM) and PBLH are derived from the 1

km horizontal resolution data. The aerosol and cloud products (AOP, ATM, COP, and CTM) and PBLH are created from the $1^*$ km horizontal resolution data. In the following section, products and algorithms related to aerosols (i.e., FM, AOP, ATM, PBLH) are mainly described. To estimate these products, the optical properties of atmospheric molecules, such as extinction and backscatter coefficients (MOP in Figure 1) are required. Here, the EarthCARE auxiliary product, containing a subset of meteorological fields from an ECMWF model (Eisinger et al. 2023) is used to compute the molecular optical propertes.



## 3 Algorithm

### 3.1 Layer identification (Feature mask)

The algorithm was developed based on retrieval methods developed using the CALIOP and ground-based lidar data (Okamoto et al. 2007 and 2008; Hagihara et al. 2010; Shimizu et al. 2004). The algorithm classifies the atmospheric layer into molecules (clear-sky), aerosol particles, or cloud particles, as well as the surface, subsurface, and layers, where the signal is fully attenuated under optically thick layers, such as clouds (Vaughan et al 2009). The copolar ($\beta_{atn.M,co}$) and crosspolar ($\beta_{atn.M,cr}$) components of the Mie attenuated backscatter coefficient and Rayleigh attenuated backscatter coefficient ($\beta_{atn.R}$) are given as ATLID L1 products and are described by the following equations with atmospheric parameters:

$$\beta_{atn,M,co} = \beta_{p,co}(z) \exp\left\{-2\int_z^{Z_{ATLID}}\left(\alpha_p(z') + \alpha_m(z')\right)dz'\right\}, \tag{1a}$$

$$\beta_{atn,M,cr} = \beta_{p,cr}(z) \exp\left\{-2\int_z^{Z_{ATLID}}\left(\alpha_p(z') + \alpha_m(z')\right)dz'\right\}, \tag{1b}$$

$$\beta_{atn,R} = \beta_m(z) \exp\left\{-2\int_z^{Z_{ATLID}}\left(\alpha_p(z') + \alpha_m(z')\right)dz'\right\}. \tag{1c}$$

$$\beta_{p,co}(z) = \beta_p(z)\frac{1}{1+\delta_p(z)}, \tag{1d}$$

$$\beta_{p,cr}(z) = \beta_p(z)\frac{\delta_p(z)}{1+\delta_p(z)}, \tag{1e}$$

$$\beta_p(z) = \frac{\alpha_p(z)}{S_p(z)}. \tag{1f}$$

where, $\alpha_p$, $S_p$, and $\delta_p$ are the extinction coefficient, lidar ratio, and depolarization ratio of particles (aerosols and clouds), respectively; $\beta_{p,co}$ and $\beta_{p,cr}$ are the copolar and cross-polar components of the particular backscatter coefficient, respectively; $\alpha_m$ and $\beta_m$ are the molecular extinction coefficient and the molecular backscatter coefficient, respectively; Z is the altitude; and $Z_{ATLID}$ is the altitude of ATLID. The Mie-attenuated backscatter coefficient ($\beta_{atn.M}$) is defined as the sum of the copolar and cross-polar components (i.e., $\beta_{atn.M} = \beta_{atn.M,co} + \beta_{atn.M,cr}$). We use $\beta_{atn.M}$ and $\beta_{atn.R}$ as the diagnostic parameters for $P_M$ and $P_R$, respectively. The parameter corresponding to the particle backscatter coefficient ($\beta_p = \beta_{p,co} + \beta_{p,cr}$) calculated from the following equation is used as the diagnostic parameter ($P_P$):

$$P_P = \beta_m(z)\beta_{atn,M}(z)/\beta_{atn,R}(z). \tag{2}$$

First, a diagnosis is made on the 0.3 km horizontal resolution data in each layer sequentially from the upper layer to the lower layer according to the following criteria.

(a) If $P_M$ and $P_R$ are not significant due to signal noise and missing data, no diagnosis is made in that layer (classified as "invalid").

(b) If $P_R$ is significant and $P_M$ is not significant, the layer is classified as "Clear-sky."

(c) $P_M$ is significant, the layer is classified as "Aerosol, Cloud, or Surface."

The significance of $P_R$ and $P_M$ is determined by the magnitude of the SNR (here, an SNR > 3 is adopted). For the layer diagnosed as "Aerosol, Cloud, or Surface," the following further diagnoses are made. For the surface detection, the criterion





is that the $P_M$ is above the threshold value set based on actual ATLID observed data and the relevant layer is below +500 m with respect to the altitude of the Digital Elevation Model (DEM) surface altitude from the EarthCARE auxiliary data to prevent misdetection due to clouds with large backscatter coefficieints. The layer below the surface is classified as "Sub-surface."

   For the cloud detection, two procedures are performed based on the cloud-mask scheme developed for the CALIOP and

shipborne lidar measurements (Hagihara et al. 2010 and Okamoto et al. 2008). First, the cloud layer is detected using the following vertical variable criteria.

$$P_M > P_{th} = 0.5\beta_{c,th} \exp(-2\tau_m)\{1 - \tanh(z - z_c)\}, \qquad (3a)$$

$$P_P > P_{th} = 0.5\beta_{c,th}\{1 - \tanh(z - z_c)\}. \qquad (3b)$$

where, $\tau_m$ is the molecular optical thickness up to an altitude ($z$) from the altitude of the ATLID. $z_c$ is given as 5 km, and

$\beta_{c,th}=10^{-5.25}$ ($\sim5.6\times10^{-6}$) [/m/sr]. The validity of these threshold values was discussed in detail by Hagihara et al. (2010) and Okamoto et al. (2008). Statistical analyses of cloud and aerosol backscatter coefficient data over several years from long-term ground-based observations by HSRL (Jin et al. 2020) and Raman lidar (Nishizawa et al. 2017) also supported the value of $\beta_{c,th}$. If the $P_R$ of the target layer is significant, the Equation 3(b) is used as the criteria; otherwise, Equation 3(a), is used.  The advantage of using $P_P$ as a diagnostic parameter is that it can eliminate attenuation due to clouds/aerosols from the ATLID to

the target layer. However, the calculation of the $P_P$ requires $\beta_{atn.R}$ in addition to $\beta_{atn.M}$, which has a disadvantage in that it cannot be diagnosed if the SNR of $\beta_{atn.R}$ is insufficient. Therefore, a hybrid method is used here, where $P_M$ is used if $P_P$ cannot be used as a diagnostic parameter. After identifying the cloud layer using these critetia, a continuity test is performed to suppress misdetection due to signal noise (Hagihara et al. 2010). The continuity test is conducted on a 5-bin horizontal and 3-bin vertical window centered on the diagnostic layer, in accordance with the continuity test method used by Hagihara et al. (2010). If more

than half (i.e. more than 8 bins) of the total 15 bins are clouds, the target layer is determined to be a "Cloud". However, if the number of cloud layers was less than half of the total, but at least one of the cloud layers was included, the layer was designated as "Unknown." For 0.3 km and 1 km resolution products, the clear-sky or aerosol layer are not diagnosed separately from the SN's point of view, and together they are classified as "Clear-sky or aeroso.l" The fully attenuated layer is diagnosed when no surface is detected; the layers below the lowest layer of "clear-sky or aerosol" or "cloud" are classified as "Fully attenuated."

Next, a diagnosis is made on the 1 km horizontal resolution data. Diagnostics similar to those performed on the 0.3 km horizontal data are performed; the layer types of "Cloud," "Clear-sky or aerosol," "Surface," " Sub-surface," "Fully attenuated," "Invalid," or "Unknown" are identified. The following method is used to identify the cloud layers. The 1 km horizontal resolution data is calculated by averaging several horizontal layers (~4 bins) of the 0.3 km horizontal resolution data. This improves the SNR, but it also causes dulling of the signal at the cloud edges, which leads to cloud layer misidentification. To

suppress this, we use the FM product estimated using the 0.3 km horizontal resolution data (i.e., FM(0.3)); if more than half of the total number of the layers of the FM (0.3) product in the target layer for the 1 km horizontal resolution data is identified as "Cloud," the target layer for the 1 km horizontal resolution data is determined to be "Cloud" (Hagihara et al. 2010). However,





if the number of cloud layers is less than half of the total, but at least one of the cloud layers is included, the target layer is designated as "Unknown." In addition, the possibility of identifying optically thin clouds or aerosols compared to the 0.3 km

data, especially at high altitudes, also arises because of the improved SNR by the horizontal averaging. Equations 4a and 4b are then applied to the 1 km horizontal resolution data with the addition of a criterion for high altitude to Equations 3a and 3b.

$$P_M > P_{th} = 0.5\beta_{c,th}\exp(-2\tau_m)\{1 - \tanh(z - z_c)\} + 0.5\beta_{c,th2}\exp(-2\tau_m)\{1 + \tanh(z - z_c)\}, \tag{4a}$$

$$P_P > P_{th} = 0.5\beta_{c,th}\{1 - \tanh(z - z_c)\} + 0.5\beta_{c,th2}\{1 + \tanh(z - z_c)\}. \tag{4b}$$

$\beta_{c,th2}$ is the threshold for identifying the layer that may be a cloud at high altitudes and is set based on actual ATLID data. If

the criteria is satisfied, the layer is marked as "Unknown," and if not, it is marked as "Aerosol or clear-sky" if the threshold is unsatisfied.

Finally, a diagnosis is made on the $1^*$ km horizontal resolution data. Diagnostics similar to those performed on the 1 km horizontal resolution data are performed; the layer types of "Cloud," "Aerosol," "Clear-sky," "Surface," "Sub-surface," "Fully attenuated," "Invalid," or "Unknown" are identified. Unlike the FM for the 1 km horizontal resolution data, the layer diagnosed

as "Aerosol or clear-sky" is classified into the "Aerosol" when the $P_M$ is significan or "Clear-sky" when the $P_M$ is not significant.

### 3.2 Aerosol optical properties

As shown in Equations 1a-1c, the POP $\alpha_p$, $\beta_p$, $\delta_p$, and $S_p$ can be directly derived using the L1 data of $\beta_{atn.M,co}$, $\beta_{atn.M,cr}$, and $\beta_{atn.R}$. The POP in the aerosol layer identified by the FM are classified as AOP. On the other hand, L1 data with a sufficient

SN are needed to derive parameters with sufficient accuracy, especially for extinction coefficient retrieval [e.g., Liu et al. 1999]. Therefore, it is essential to average and/or smooth the measured signals to improve the SNR.

The L1 data may have negative values owing to large signal noise. In this situation, POP cannot be retrieved directly from the L1 data. Therefore, we simultaneously optimize the vertical profiles of the POP to the L1 data with smoothness constraints for the vertical profiles of the POP using the optimal estimation technique developed by Kudo et al. [2016]. The cost function

is defined as follows:

$$F(\boldsymbol{x}) = \sum_i \frac{\left\{ln\left(\beta_{atn.M,co}^{obs}(z_i) - \beta_{atn.M,co}^{min}\right) - ln\left(\beta_{atn.M,co}^{cal}(z_i) - \beta_{atn.M,co}^{min}\right)\right\}^2}{w_{atn.M,co}^2(z_i)}$$

$$+\sum_i \frac{\left\{ln\left(\beta_{atn.M,cr}^{obs}(z_i) - \beta_{atn.M,cr}^{min}\right) - ln\left(\beta_{atn.M,cr}^{cal}(z_i) - \beta_{atn.M,cr}^{min}\right)\right\}^2}{w_{atn.M,cr}^2(z_i)}$$

$$+\sum_i \frac{\left\{ln\left(\beta_{atn.M,R}^{obs}(z_i) - \beta_{atn.M,R}^{min}\right) - ln\left(\beta_{atn.M,R}^{cal}(z_i) - \beta_{atn.M,R}^{min}\right)\right\}^2}{w_{atn.M,R}^2(z_i)}$$

$$+\sum_i \left\{ln\left(\alpha_p(z_i)\right) - ln\left(-\alpha_p(z_{i+1})\right)\right\}^2$$





$$+ \sum_i \left\{ ln\left(S_p(z_i)\right) - ln\left(-S_p(z_{i+1})\right)\right\}^2$$

$$+ \sum_i \left\{ ln\left(\delta_p(z_i)\right) - ln\left(-\delta_p(z_{i+1})\right)\right\}^2, \qquad (5)$$

where $z_i$ is $i$-th altitude, "obs" indicates the measurements, "cal" indicates the values calculated from the POP by Equations 1a-1f. "min" indicates the possible minimum value of the measurements, and $w$ is measurement uncertainties. A logarithmic

transformation is applied to the measured and calculated values. Because the measured lidar signals have large dynamic ranges of more than two orders of magnitude, the terms in Equation 5, with small measured and calculated values, are ignored, and the POP at that altitude cannot be optimized. Logarithmic transformation reduces the differences between the digits of each term in Equation 5. We subtract the possible minimum values from the measured and calculated values because logarithmic transformation cannot be applied to negative values. The fourth, fifth, and sixth terms in Equation 5 are the smoothness

constraints for the vertical profiles of the POP. The smoothness of the vertical POP profiles is obtained by minimizing the differences in the POP at two adjacent altitudes.

The optimized POP, which minimizes the cost function, is searched by the iteration of $ln(x_{i+1}) = ln(x_i) + \gamma \Delta x_i$ in $ln(x)$ space, where the state vector $x$ is comprised of the POP. The vector $\Delta x_i$ at the i-th step is determined by the Gauss-Newton method, and the scalar $\gamma$ is determined by a line search method with Armijo rule. The convergence criterion for the iteration

is that the difference between $F(x_i)$ and $F(x_{i+1})$ should be smaller than a given threshold.

**3.3 Aerosol type classification (Target mask)**

The algorithm uses the derived FM and AOP products. The algorithm classifies aerosol type for each layer that the FM scheme identifies as "Aerosol." The algorithm uses differences in the light absorption and polarization properties of aerosol types; $S_a$ strongly reflects the light absorption of aerosols, while $\delta_a$ strongly reflects the polarization of aerosols. Therefore, it

is necessary to determine the aerosol types and their optical models. The optical properties and size distributions of aerosols are observed by the AErosol RObotic NETwork (AERONET) sun/sky photometer, and AERONET observation sites are located worldwide on all continents (e.g. Holben et al. 1998; Giles et al. 2019). In the retrieval of aerosol extinction in CALIPSO version 2 and 3 products, $S_a$ of three aerosol types calculated by using size distributions and refractive indices of the clusters grouped by the cluster analysis of the AERONET dataset were used (Omar et al. 2005; Omar et al. 2009). Based

on this assumption, we also perform a cluster analysis of the aerosol optical properties estimated from the AERONET data and determined the aerosol optical properties at 355 nm. In this study, aerosol particles are classified as six aerosol types ("Smoke," "Pollution," "Marine," "Pristine," "Dust," and "Dusty mixture") with "Unknown" in Figure 2.





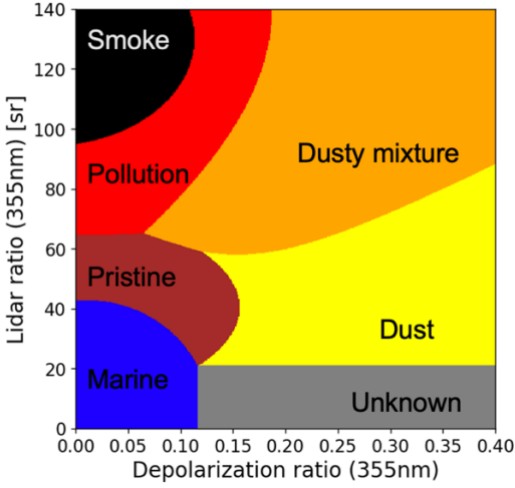

Figure 2. Lidar ratio – depolarization ratio diagram for classifying aerosol types.


The optical properties of the four aerosol types (smoke, pollution, marine, and pristine) are determined using cluster analysis of the AERONET data. We perform a cluster analysis of the AERONET southern Africa sites for smoke type, that of Chinese sites for pollution type, and that of island sites for marine and pristine types. First, $\delta_a$ and $S_a$ at 355, 532, and 1064 nm, which are usually observed by the HSRL and Raman lidar measurements, are calculated using the refractive indices at 440, 675, 870,

and 1020 nm, size distributions of fine and coarse modes, and the sphericity of scattering light derived from the AERONET inversion algorithm (Dubovik et al., 2006) for each AERONET data sample. The refractive indices at 532 nm are interpolated and those at 355 and 1064 nm are extrapolated using those of AERONET product from 440 to 1020 nm. The non-spherical particle shape is assumed to be the AERONET spheroid model (Dubovik et al., 2006). Next, we adopt the fuzzy c-means method to conduct a cluster analysis. This method is based on minimizing the following objective function:

$$J = \sum_{i=1}^{N}\sum_{k=1}^{K} g_{ik}^2 \|x_i - c_k\|^2, \qquad (6)$$

where $g_{ik}$ is the degree of membership of $x_i$ to the $k$-th cluster, $x_i$ is the observed data, $c_k$ is the center of the $k$-th cluster. Partitioning is conducted through an iterative optimization of the $J$ with updates of $g_{ik}$ and $c_k$. Based on this cluster analysis, the center of cluster $c_k$ is assumed to be the representative value of the selected cluster parameter. We select 12 parameters used in the cluster analysis to classify the aerosol properties. These parameters are $\delta_a$ and $S_a$ at 355, 532, and 1064 nm and the

imaginary part of the refractive index and fine mode fraction (FMF) to the total (fine+coarse) AOT at 440, 675, and 870 nm. Finally, we define the characteristic results of the cluster as the optical properties of the aerosol type. The cluster, which is fine-mode dominated and contains the most light-absorbing aerosols, is defined as the smoke type in the southern Africa analysis and is defined as the pollution type in the Chinese analysis. The marine and pristine types are defined based on islands analysis. The marine type has the largest particle size with an SSA of 0.98. The pristine type has a smaller particle size than

the marine type, with the SSA of 0.98.





The difference in $\delta_a$ and $S_a$ of non-spherical dust particles between observations and theoretical calculations remain large (Tesche et al., 2019), so that $\delta_a$ and $S_a$ of the dust type at 355 nm are referred to as the averaged values of the Raman lidar observations in Morocco (Freudenthaler et al. 2009; Tesche et al. 2009), Germany (Wiegner et al. 2011), and Tajikistan (Hofer et al. 2017). The dusty mixture type is defined as a mixture of dust and smoke in ratios of 0.65 and 0.35, respectively. Aerosol particles with high depolarization ratios and low lidar ratios are rarely observed; therefore, the unknown aerosol type is defined as aerosols with $\delta_a > 0.12$ and $S_a < 21$ sr, which are determined by the values of the intersection of the border line between dust and pristine types and that between marine and pristine types, as shown in Figure 2.

**3.4 Planetary boundary layer height**

The PBL is the lowest atmospheric layer in which the temperature, wind, and water vapor mixing ratio are influenced by the Earth's surface, and its top height is an important parameter for understanding air quality. Because of the temperature inversion between the PBL and free troposphere, aerosols emitted from the surface are trapped within the PBL. As a result, aerosol concentrations in the PBL are higher than those in the free troposphere. The transition point of the concentration gap is characterized by the PBLH, which can be detected by aerosol lidar using the gradient method (Lammert and Bösenberg, 2006), the wavelet covariance transform (WCT) method (Brooks, 2003), or other methods (e.g., the standard deviation method by Menut et al., 1999). Among these, the WCT method is less affected by noise (Qu et al., 2017), and is promising for the spaceborne lidar (Kim et al., 2021).

For PBLH detection, lidar signals at wavelength of 532 or 1064 nm are mostly used; however, the ATLID wavelength is 355 nm. Because the Rayleigh scattering at 355 nm is relatively large (approximately five times larger than that at 532 nm), the difference in the backscatter signals for the PBL and the free troposphere can be small. Moreover, because the spaceborne lidar signals attenuate downward, attenuated backscatter in the PBL results in a small; therefore, the detection of PBLH is more challenging than with ground-based lidars even when the WCT method is applied (Kim et al., 2021). In this study, instead of using attenuated backscatter signals as the input of the WCT method, the backscattering ratio (BR), which can be calculated as the ratio of Mie-attenuated backscatter to Rayleigh-attenuated backscatter from the ATLID L1 data (i.e., $BR = \beta_{atn.M} / \beta_{atn.R}$), is used to remove the effect of signal attenuation. Also, Rayleigh scattering component is eliminated by calculating BR minus 1 (i.e., $BR' = BR - 1$) to highlight aerosol scattering intensity in the PBL.

The BR' and FM are used as input data. Here, the target altitude (height above the surface) is set in between $z_{min}$ and $z_{max}$, which are 0.1 and 5 km, respectively. If the target point is classified as a cloud in the FM at the target altitude, it is excluded from the PBLH detection. Then WCT is the calculated using the following equation:

$$WCT(a,b) = \frac{1}{a} \int_{z_{min}}^{z_{max}} BR'(z) h\left(\frac{z-b}{a}\right) dz. \tag{7}$$

where $a$ and $b$ are the dilation and the centered location of the Haar function $h$, which is defined as:



$$h\left(\frac{z-b}{a}\right) = \begin{cases} +1: b - \frac{a}{2} \le z < b, \\ -1: b \le z \le b + \frac{a}{2}, \\ \quad 0: \text{elswere.} \end{cases} \tag{8}$$

In Equation 7, BR′ is normalized to 1.0 for the height from the surface to 1 km to reduce the dependency of aerosol concentrations. When the WCT exceeds a threshold value, the first WCT peak from $z_{min}$ is determined as the PBLH. The threshold and dilation width are determined from a simulated BR′ profile based on ground-based HSRL data (Jin et al., 2020)

with random noise according to the errors in the ATLID attenuated backscatter reported by do Carmo et al., (2021), and are set to 0.2 and 1.0 km, respectively.

## 4 Results and Discussion

To demonstrate the products and the performance of the algorithm, the ATLID L1 data (i.e., Equations 1a-1c) simulated using the Joint Simulator for Satellite Sensors (Joint-Simulator) (Hashino et al. 2013, Satoh et al. 2016, Row et al. 2023) were

used. Cloud (and precipitation) distributions simulated using the Nonhydorstatic Icosahedral Atmospheric Model (NICAM; Satoh et al., 2014) and aerosol distributions simulated using NICAM Spectral Radiation Transport Model for Aerosol Species (NICAM-SPRINTARS; Takemura et al., 2000) were used as the input data. Signal noises such as shot noise, dark noise, and CCD read-out noise expected from the ATLID system were evaluated and added as gaussian random noise.

The ATLID L1 data for the cloud-predominant case simulated along the satellite path using Joint-Simulator are shown in

Figure 3. The cloud and aerosol fields used for the signal simulation are also shown in Figure 4. Clouds are present from the surface to the altitude of 15 km. The SNR of the Mie copolar- and crosspolar-attenuated backscatter coefficients for the cloud layers are generally greater than 5 for 0.3 km horizontal resolution data; the SNR for 1 km horizontal resolution data is approximately twice than that for 0.3 km horizontal resolution data. The FM algorithm (Section 3.1) was applied to the simulated L1 data (Figure 5). To assess the effect of signal noise on the estimates, the algorithm was applied to L1 data with

or without signal noise, and the agreement between them was evaluated. The results for the 0.3 km and 1 km horizontal resolution data showed generally good agreement for each layer type, such as "cloud" and "clear-sky or aerosol." For the 0.3 km horizontal resolution data, the total number of layers for the cloud type was 226,000, of which the number of misidentification was 24,000, corresponding to a 11% relative error. For the 1 km horizontal resolution data, the misidentification of the cloud layers was 9%. For the clear-sky or aerosol type, the misidentification reached to 41% for the

0.3 km horizontal resolution data; however, it improved to 5% for the 1 km horizontal resolution data. Many misidentification were noted at the edges of the cloud layers, highliting the need for improvement is this aspect in future studies.





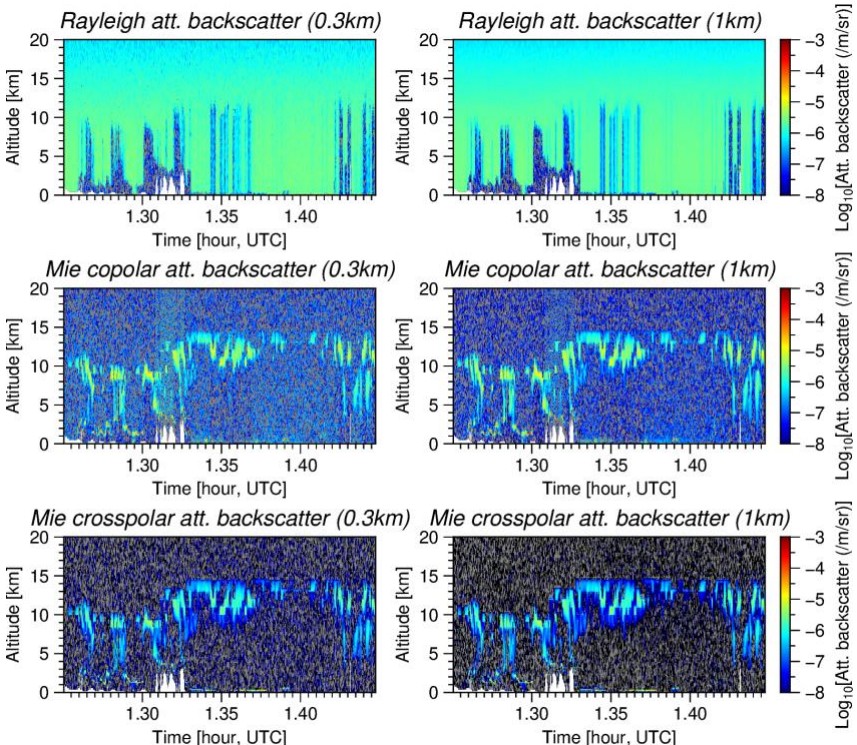

Figure 3. Rayleigh-, Mie copular-, and Mie crosspolar-attenuated backscatter coefficients at 355 nm for cloud predominant scenes simulated using Joint-Simulator. The left/right figure shows the 0.3/1 km horizontal resolution data.

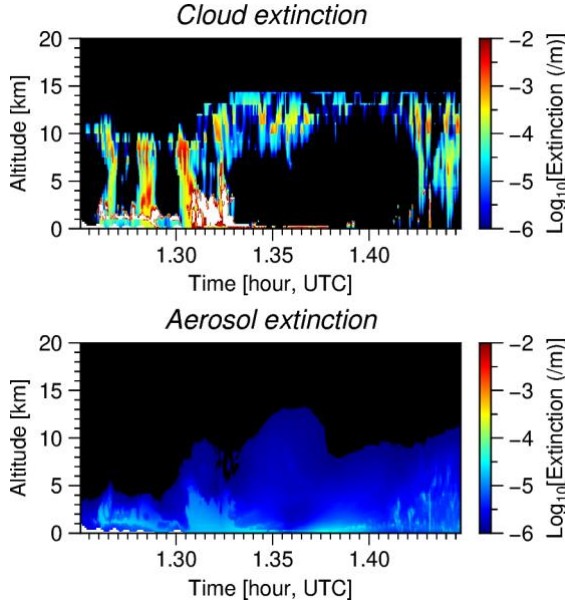


Figure 4. Extinction coefficient at 355 nm of clouds and aerosols used for the simulations for cloud predominant case (Figure 3).

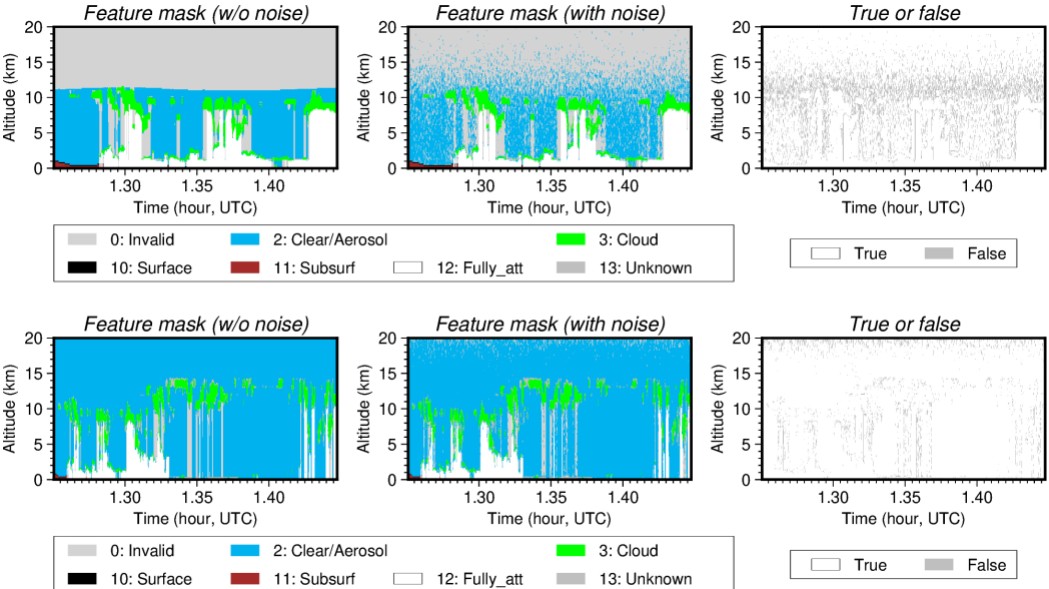

Figure 5. Feature mask products estimated for data with/without signal noise for cloud predominant scenes (Figure 3) and its true or false values. The upper/lower figures show the results for the 0.3/1 km horizontal resolution data.

The results of the aerosol layer identification are shown in Figure 7. Here, we used the the L1 data simulated for the aerosol-predominant case, where the dust layer is widely suspended over an altitude range of 3-20 km (Figure 6), and the FM algorithm was applied to the L1 data with or without noise. The results for the data with signal noise are generally in good agreement with those for the data without signal noise. The total number of aerosol layers was approximately 280,000 and the number of misidentified layers was 30,000, resulting in 11% misidentification. As in the cloud-predominant case (Figure 5), much of the misidentification occurred at the edges of the aerosol layers, which is also an aspect that can be improved for aerosol identification.



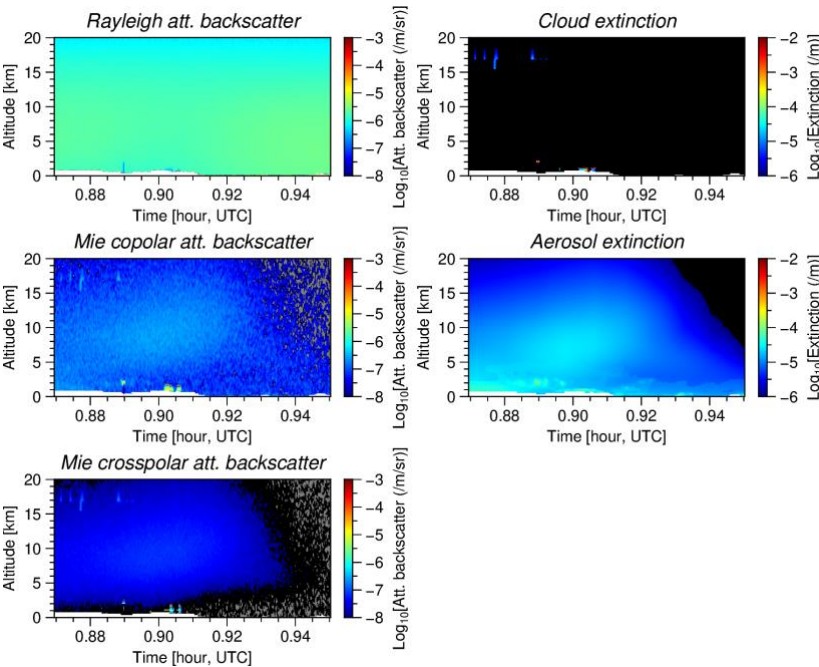

Figure 6. Rayleigh-, Mie copular-, and Mie crosspolar-attenuated backscatter coefficients at 355 nm for aerosol predominant scenes simulated using Joint-Simulator and extinction coefficients at 355 nm of aerosols and clouds. The simulations shown used 1* km horizontal resolution data.

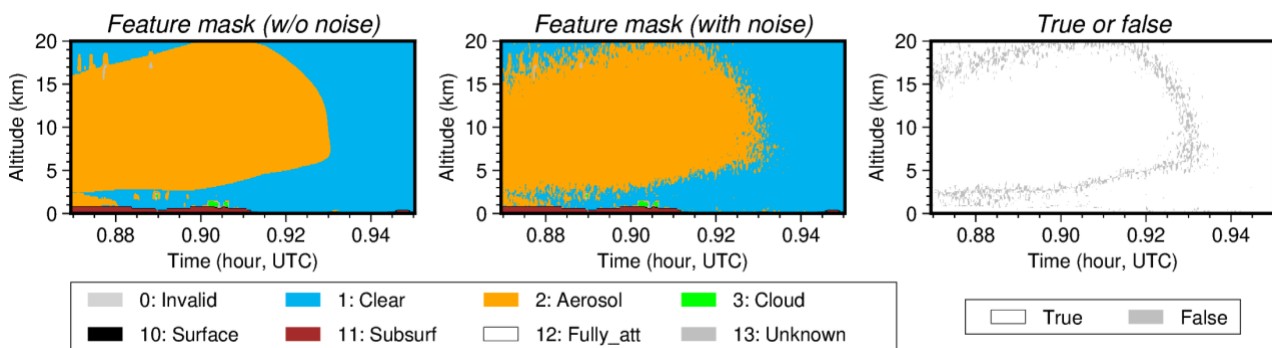

Figure 7. Feature mask products estimated for data with/without signal noise for aerosol predominant cases (Figure 6) and its true or false values.

The results of the AOP retrieval are shown in Figure 8. The AOP retrieval algorithm was applied to the aerosol-predominant case, as shown in Figure 6. The SNRs of the L1 data for the dust layer generally ranged from 5 to 20 for $1^*$ km horizontal resolution data. The estimated backscatter coefficients and depolarization ratios generally agreed well with the actual values, which were the aerosol optical properties used in the L1 simulation. The mean of retrieved aerosol backscatter coefficient was





$3.26 \times 10^{-7}$ m$^{-1}$sr$^{-1}$. The true value was $3.18 \times 10^{-7}$ m$^{-1}$sr$^{-1}$ and the mean error (ME: retrieval − truth) was $0.08 \times 10^{-7}$ m$^{-1}$sr$^{-1}$, corresponding to a relative error of 2%. The root mean square error (RMSE) was $1.12 \times 10^{-7}$ m$^{-1}$sr$^{-1}$, corresponding to a relative

error of 34%. For the depolarization ratio, the means for the retrievals and true values were 0.27 and 0.26, respectively; ME = 0.01 (4%) and RMSE = 0.07 (27%). The figure also shows that the extinction coefficient and lidar ratio were more strongly affected by the signal noise than the backscatter coefficient and depolarization ratio. For the extinction coefficient, the means for the retrieval and true values were $1.32 \times 10^{-5}$ m$^{-1}$ and $1.35 \times 10^{-5}$ m$^{-1}$, respectively; ME = $-0.03$ m$^{-1}$ (2%) and RMSE = $1.05 \times 10^{-5}$ m$^{-1}$ (78%). For the lidar ratio, the means of the retrieval and true values were 41 sr$^{-1}$, respectively; ME = 0 sr$^{-1}$ (0%) and

RMSE = 25 sr$^{-1}$ (61fa). As with the backscatter coefficient and depolarization ratio, there was no significant bias error (ME) for the extinction coefficient and lidar ratio; however, the variation (RMSE) was relatively large owing to signal noise. Notably, the aerosol concentration used in this analysis was relatively low (i.e., $1.35 \times 10^{-5}$ m$^{-1}$ on an average), and the RMSE value of the extinction coefficient is not large (i.e., $1.05 \times 10^{-5}$ m$^{-1}$). For cases with higher aerosol concentrations, it can be expected that the relative error of the extinction coefficient and the RMSE of the lidar ratio decrease with an increase in the SNR of the

Mie-attenuated backscatter coefficient (e.g., Nishizawa et al. 2017).

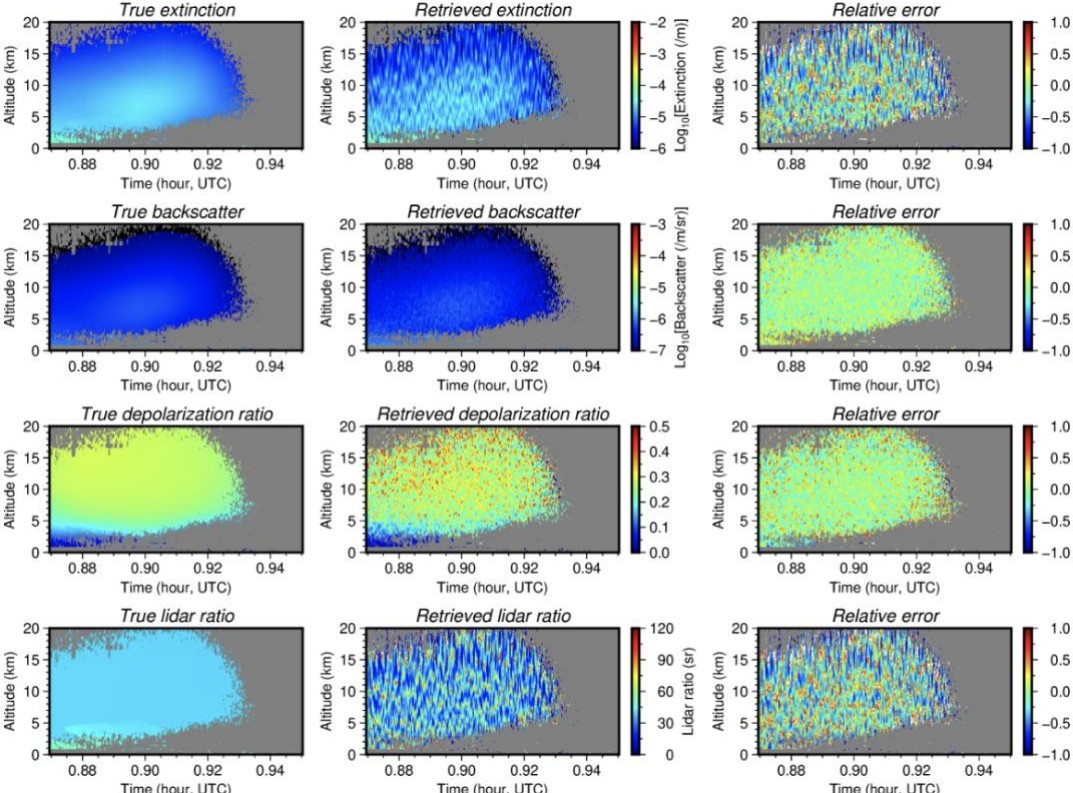

Figure 8. Extinction coefficient, backscatter coefficient, depolarization ratio, and lidar ratio of aerosols estimated for aerosol predominant cases (Figure 6) and its relative error.





In the actual analysis using the ATLID data, the optical properties of clouds and aerosols are simultaneously estimated using
the developed POP retrieval algorithm (Section 3.2). To estimate the optical properties of optically thick scatterers such as clouds, it is essential to compute L1 data considering multiple scattering. The multiple scattering has been considered in CALIPSO and ground-based lidar analysis by introducing η-factor (e.g., Chen et al. 2002, Young 2013, Cairo et al. 2021), and thus the η-factor will be introduced into this algorithm to estimate the optical properties of clouds. Sato et al. (2018) developed a practical model to determine the time-dependent lidar attenuated backscatter coefficient, in which an analytical expression
for the high-order phase function was implemented to reduce computational cost; furthermore, Sato et al. (2019) developed a vectorized physical model (VPM) which is a physical model extended with a polarization function, to analyze the observed depolarization ratio due to multiple scattering from water clouds. The introduction of these physical models is a promising as a more advanced and accurate approach in estimating COP, considering multiple scattering from clouds.

The results of the aerosol-type classification are shown in Figure 9. The algorithm was applied to the L1 data with and
without signal noise for the aerosol-predominant case, as shown in Figure 6. For the dust layer, there appeared some misclassification (e.g., dusty-mixture) and "unknown" layers; however, there was a fair amount of agreement. The total number of dust layers was approximately 225,000 and the number of misidentified layers was 84,000, resulting in a 37% misidentification. In addition, pristine-type aerosol layers were also found below an altitude of 4 km; the total number of pristine layers was approximately 13,000 and the number of misidentified layers was 9,700, resulting in 74% misidentification.
Aerosol typing was classified using a two-dimensional diagram of the lidar and depolarization ratios (Figure 2). For pristine particles categorized as spherical, the lidar ratio is a key parameter for identification; thus, an accurate estimation of the lidar ratio is required. For cases with higher aerosol concentrations, it can be expected that the misclassification of aerosol types decreases with an decrease of the retrieval error of the lidar ratio, as discussed in the AOP retrieval (Section 4).

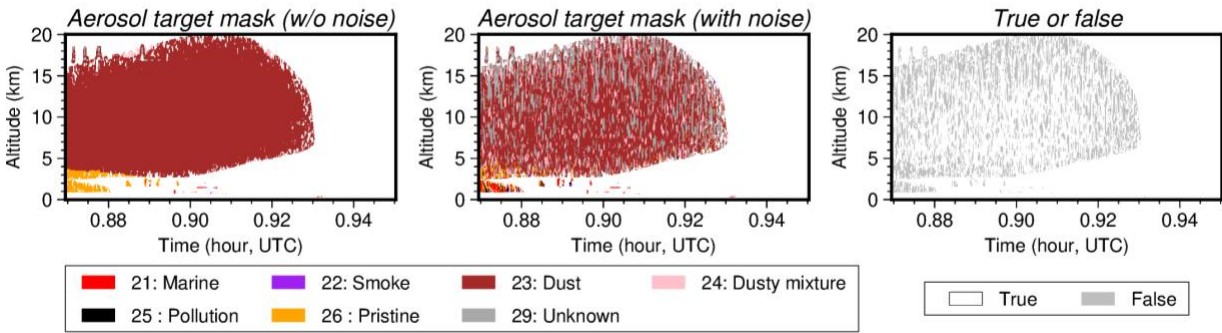

Figure 9. Aerosol target mask products estimated for data with/without signal noise for aerosol predominant scenes (Figure 6) and its true or false values.

In the actual analysis using ATLID data, cloud types will be classified together with aerosol types. Multiple cloud particle types such as warm water, supercooled water, two-dimensional ice, and three-dimensional ice, can be identified by applying a





two-dimensional diagrammatic method of signal attenuation (or extinction coefficient) and depolarization ratio developed for

CALIOP cloud type classification (Yoshida et al. 2010).

Figure 10 shows an example of the PBLH detection for the L1 data simulated using Joint-Simulator, where the PBLH was

approximately 2.1 km. The WCT was calculated using a Haar function with a dilation width of 1.0 km, and a WCT peak

exceeding a threshold of 0.2, derived at 2.1 km. When signal noise was added to the L1 data (dotted line in Figure 10), another

WCT peak at ~1 km appeared because of signal fluctuation; however, the peak was below the threshold. In this case, the

estimated PBLH with noise agreed well with that without noise. In actual observations, the PBLH detection may be difficult

for cases of noisy data, low aerosol concentrations, and the existence of residual layers at night because the gap of BR′ for the

PBL and free troposphere may be small.

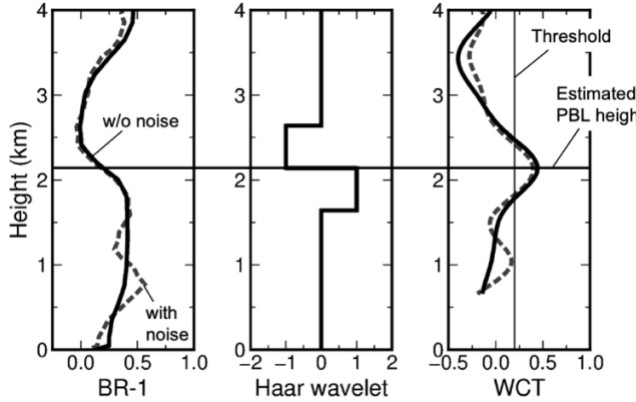


Figure 10. Vertical profiles of (left) backscattering ratio (BR) minus 1 for the simulated ATLID L1 data, (center) Haar function,

and (right) wavelet covariance transform (WCT).

**5 Conclusion**

We developed algorithms to produce ATLID L2 products for aerosols and clouds using ATLID L1 data. The algorithms

retrieve the following products: (1) feature mask; (2) particle optical properties such as extinction coefficient, backscatter

coefficient, depolarization ratio, and lidar ratio at 355 nm; (3) target mask; and (4) planetary boundary layer height. The

algorithm performance was demonstrated using ATLID L1 data with realistic signal noise calculated based on the ATLID

specification and simulated for aerosol- or cloud-predominant cases by the Joint-Simulator. The main findings of this

simulation study are as follows: (1) the misidentification of the aerosol and cloud layers by the feature mask algorithm was

relatively low (approximately 10%). (2) The retrieval errors of the AOP were $0.08 \times 10^{-7} \pm 1.12 \times 10^{-7}\,\mathrm{m^{-1}sr^{-1}}$ ($2 \pm 34\%$ relative

error) for the backscatter coefficient and $0.01 \pm 0.07$ ($4 \pm 27\%$ relative error) for the depolarization ratio of aerosols; the relative

errors of the extinction coefficient and lidar ratio were worse than those of the backscatter coefficient and depolarization ratio.

(3) The aerosol-type classification generally performed well, with 37% misclassifications for the dust type, although there





were more misclassifications for the pristine type. (4) PBLH retrieval using the WCT method with ATLID L1 data was feasible.
These results indicate that the algorithm's capability to provide valuable insights into the global distribution of aerosols and clouds, facilitating assessments of their climate impact through atmospheric radiation processes.

The aerosol and cloud products estimated using the algorithms developed in this study will be released as JAXA's L2 standard products. Along with the development of the standard algorithms, we have been developing an algorithm to estimate the extinction coefficients of dust, sea salt, carbonaceous (light-absorbing particles), and water-soluble aerosols at 355nm
using the difference in the depolarization and light absorption propertes of each aerosol component from the L2 standard products and ATLID L1 data. Furthermore, we have also been developing an ATLID-MSI synergy algorithm to retrieve the vertical mean mode-radii of dust and fine-mode aerosols, and the extinction coefficients of the abovementioned four aerosol components. These algorithms have been developed based on the aerosol component retrieval algorithms that have been developed for the analysis of CALIOP and ground-based lidar data. (Kudo et al. 2023; Nishizawa et al. 2007, 2008, 2011,
2017). The aerosol component products derived by these algorithms using ATLID and MSI data will be released as JAXA's L2 research products, which are recommended for production to enhance the scientific value of the EarthCARE mission.

The validation of these aerosol and cloud products is essential, and JAXA and ESA plan to conduct validation using various platforms, such as airborne, shipborne, and ground-based measurements, and various observation instruments, including lidars. During this validation study, the improvement of the L2 standard algorithms and the research algorithms mentioned above,
including the verification and improvement of various assumptions (e.g., optical and microphysical models of aerosols and clouds), and the criteria used in the algorithms, and the accuracy of the products, will be guaranteed.

**Author contributions.**

TN developed the algorithm concept, a feature mask scheme, and managed the project. RK developed a scheme for retrieving
the particle optical properties. EO developed a scheme to classify the aerosol types. YJ developed a scheme for estimating the planetary boundary layer height. AH, EO, YJ, and RK constructed code integrating the individual algorithms and performed an error analysis using the simulated data. NS, KS, and HO generated ideas that contributed to improvements in the retrieval schemes. TN prepared the paper with contributions from all the co-authors.

**Competing interests.**

The contact author has declared that none of the authors has any competing interests.

**Special issue statement.**

This article is part of the special issue "EarthCARE Level 2 algorithms and data products." However, this is not associated
with conferences.

**Acknowledgments.**





The authors would like to thank D. Donovan and G. J. van Zadelhoff for their support and advice regarding ATLID signal-noise calculations. We thank the principal investigators and staff at the AERONET sites used in this study for maintaining the stations. We thank the members of the JAXA EarthCARE Science Team and the Joint-Simulator project. We would like to thank Editage (www.editage.jp) for English language editing.

**Financial support.**

This research was supported by the EarthCARE satellite study commissioned by the Japan Aerospace Exploration Agency (grant no. 23RT000223) and a Grant-in-Aid for Scientific Research (KAKENHI) (project numbers: JP18KK0289, JP17H06139, JPS17H04477, JP15H01728, JP15H02808, and JP25220101).

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
