# Peer review of "Algorithms to retrieve aerosol optical properties using lidar measurements on board the EarthCARE satellite"

_Atmospheric Measurement Techniques, 2024_

## Author Comment (AC1)

**Reviewer 1**

**General Remarks**
This paper provides a useful concise overview of the JAXA EarthCARE lidar products and will be a useful reference for the community. I recommend publication. There are, however, several mainly editorial issues that need to be addressed.

**Title:**
This paper describes several algorithms. I suggest changing "Algorithm" to "Algorithms" in the title.
=>Thank you for pointing this out. We have made the correction as noted.

**Abstract:**
The abstract is, in general, awkward to read. It should be re-written
For example:
-EarthCARE should be introduced. "ATLID (Atmospheric lidar)" does not mean much to many readers by itself without more context.
-"optimization method using the Gauss-Newton method combined.". The numerical methods used in the optimization procedure are not interesting enough to be included in the abstract !    It would be more suitable to mention what is being optimized (e.g. have you implemented an optimal estimation type procedure ? )
-"algorithm's performance". Since more than one algorithm is being treated, the phrase "The performance of the various algorithms was evaluated"
=>Thank you for your valuable comments. We have revised the abstract in line with your comments, focusing on introducing the description of EarthCARE, simplifying the description of optimization, and the description of algorithm performance. The revised abstract is as follows.

"The Earth Cloud Aerosol and Radiation Explorer (EarthCARE) is a joint Japanese-European satellite observation mission for understanding the interaction between cloud, aerosol, and radiation processes and improving the accuracy of climate change predictions. The EarthCARE satellite was equipped with four sensors, a 355 nm high-spectral-resolution lidar with depolarization measurement capability (ATLID) as well as a cloud profiling radar, a multi-spectral imager, and a broadband radiometer, to observe the global distribution of clouds, aerosols, and radiation. In this study, we have developed algorithms to produce ATLID    Level 2 aerosol products using ATLID Level 1 data. The algorithms estimated the following four products: (1) Layer identifiers such as aerosols, clouds, clear-skies, or surfaces were estimated by the combined use of vertically variable criteria and spatial continuity methods developed for the CALIOP (Cloud-Aerosol Lidar and Infrared Pathfinder Satellite Observation) analysis. (2) Aerosol optical properties such as extinction coefficient, backscatter coefficient, depolarization ratio, and lidar ratio at 355 nm were optimized to ATLID L1 data by the method of maximum likelihood. (3) Six aerosol types, namely smoke, pollution, marine, pristine, dusty-mixture, and dust were identified based on a two-dimensional diagram of the lidar ratio and depolarization ratio at 355 nm developed by cluster-analysis using the AERONET (AErosol RObotic NETwork) dataset with ground-based lidar data. (4) The planetary boundary layer height was determined using the improved wavelet covariance transform method for the ATLID analysis. The performance of various algorithms was evaluated using pseudo ATLID Level 1 data generated by Joint-Simulator (Joint Simulator for Satellite Sensors), which incorporates aerosol and cloud distributions simulated by numerical models. Results from applying the algorithms to the pseudo ATLID Level 1 data with realistic signal noise added for aerosol or cloud predominant cases revealed: (1) misidentification of aerosol and cloud layers was relatively low, approximately 10%; (2) the retrieval errors of aerosol optical properties were $0.08\times10^{-7} \pm 1.12\times10^{-7}$ m$^{-1}$sr$^{-1}$ ($2 \pm 34\%$ in relative

error) for backscatter coefficient and 0.01±0.07 (4 ± 27% in relative error) for depolarization ratio; (3) aerosol type classification was generally performed well. These results indicate that the algorithm's capability to provide valuable insights into the global distribution of aerosols and clouds, facilitating assessments of their climate impact through atmospheric radiation processes."

**1 Introduction:**

Line 53: "...extraction of the component parallel to the laser polarization (co-polar component)..."
Line 54 delete "published" ═ > "calibrated"
Line 55 "understandings" ══> "understanding"
Line 77: "...generate JAXA L2 products using.."
Line 79: "..cloud properly estimation.."
=> We have corrected it as noted.

**2 Algorithm flow and products**

Line 85 : "Initially, the algorithm" ... which algorithm ? I guess this is referring to the "signal smoothing" step in Fig 1. ? Please re-work this sentence.
=>The text has been revised and the corresponding section (smoothing) in Figure 1 has been corrected for greater clarity as follows.

"First, to improve signal quality, the algorithm reduces the signal noise using a discrete wavelet transform (DWT) (Fang and Huang, 2004)."

[Figure]

Figure 1. ATLID L2 products and the flow of algorithms.

Line 114 : "..ECMWF forecast model.."
=>We have corrected according to your comment.

**3 Algorithm**
**Layer Identification**
Line 115: "Algorithm" ══> Algorithms
=> We have corrected "Algorithm" to "Algorithms"

Line 129 : "..and linear depolarization ratio.."

=> We have corrected as indicated

Line 133-134: Pm and Pr are not defined ! Or does e.g. Pm=beta_atn_M ? and Pr=beta_atn_R ? If this is the case, it is unnecessary and confusing in the description. Please adjust the subsequent description and Equations 3 and 4 to use beta_atn_M etc..

=> As you may have guessed, section 3.1 has been modified to remove PM and PR and write in βatn.M and βatn,R.

Line 168: "SN's" ? Do you mean "..are not identified separately using the SNR" ?

=> We modified "SN's" to "SNR's".

**Aerosol optical properties**

Can the authors give an indication of how computationally demanding their approach is ? i.e. how long (and on what type of computing system) does it take to profile a frame of Atlid data ?

=> The computational time for the retrieval of aerosol optical properties in a frame of ATLID L1 data (about 5000 profiles) is less than 15 minutes on Linux platform.

Line 193: If I understand correctly, the forward model being employed is described by Eq1 1a-1c. Is there any account of lidar multiple scattering ?

=> The multiple scattering is not considered in the retrieval of aerosols. However, the actual analysis (as described in the manuscript) estimates the optical properties of the cloud along with the aerosol, so the cloud estimation takes multiple scattering into account. The algorithm already implements the method using the η-factor, but since we are dealing with aerosols in this study, we performed various estimations assuming $\eta = 1$ (no multiple scattering).

Line 198: "...optimize the vertical profiles of the POP to the L1 data...". I am not sure what is meant here ? Maybe the authors mean to say that "...optimize the difference between the observed and forward modeled L1 profiles based on the POP profiles"

=> Thank you for your comment. The sentence was modified to "Therefore, we simultaneously estimated the vertical profiles of the POP from the L1 data by the method of maximum likelihood with a priori smoothness constraints for the vertical profiles of the POP. The state vector x, which comprises of alpha(zi), δ(zi), and S(zi) at altitudes zi, is optimized to the L1 data by minimizing the following cost function:".

Please describe how the w terms in Eq 5 are determined ? I guess they are the log uncertainties based on the (linear) error estimations in the alpha and beta determinations ?

=> the w terms are determined from the measurement error of the L1 data. An explanation has been added to the revised manuscript to clarify this point.

Are the w terms also adjusted to control the "smoothness" of the results ?

=> We added the w terms for the smoothness constraints to the Equation 5 in the revised manuscript. The smoothness constraints are controlled by the w values.

Esq. (5). It looks like there are extra "-" signs in the last three terms of the equation. e.g. -ln(-alpha_p(z_(i+1)).
=> Thank you for pointing that out. We have removed it.

Line 199: Is this really and "optimal estimation technique" ? The method looks like some sort of forward modeling approach coupled with smoothing constraints but I do not think it can be described as an "optimal estimation" technique. I.E. optimal estimation involve some sort of a prior constraint, not smoothness constraints.
=> Yes, we think the optimal estimation technique. In this study, the cost function is defined based on the method of maximum likelihood, and the state vector is optimized by minimizing the cost function.

Line 217-220: Here the authors (finally) introduce the state-vector (x). The discussion would be much easier to follow if this was done explicitly at the beginning of this sub-section.
=> Thank you for your comment. We moved the introduction of the state vector to the beginning of this subsection in the revised manuscript.

Line 219 : Provide a reference for the "Armijo" rule.
=>The reference was added to the revised manuscript.
Nocedal, J. and Wright, S. J.: Numerical optimization, 2nd edition, Springer Series in Operations Research and Financial Engineering, 664 pp., Springer Science+Business Media, LCC, New York, 2006.

**Aerosol type classification (Target mask).**
Line 244: Please provide a reference for the "fuzzy c-means method"
=> The following reference has been added.
Bezdek, J. C.: Pattern recognition with fuzzy objective function algorithms, Plenum Press, 1981.
Dunn, J. C.: A fuzzy relative of the ISODATA process and its use in detecting compact well-separated clusters, J. of Cybernetics, 3, 3, 32-57, 1973.

**PBL height**
Line 264 : "...ratio are directly influenced...."
=> We have corrected as indicated.

Line 275 : "..results in a small;.." ?    Small what ? This sentence seems corrupt.
=> To clarify this sentence, the following modifications have been made, including around the pertinent text.
"Because the Rayleigh scattering at 355 nm is relatively large (approximately five times larger than that at 532 nm), the difference in the backscatter signals for the PBL and the free troposphere (FT) can be small. This larger Rayleigh scattering at 355nm also produces greater signal attenuation, resulting in the lower signal difference between the PBL and the FT. It should be noted that the signal attenuation due to Rayleigh scattering near the top of the PBL is larger for spaceborne lidar observations than for ground-based lidar observations. Thus, the detection of PBLH by spaceborne lidar observation at 355nm is more challenging than in the past, even when the WCT method is applied (Kim et al., 2021)."

**Results and Discussion**
Line 293 : "..algorithm,.." ==> "..algorithms,..".
=> We have corrected it as indicated.

Line 298 : Were lidar multiple-scattering (MS) effects included in the simulations ? MS is described later in lines 359-365 but it is unclear(to me) if these effects were incorporated into the simulations used in this paper.

=> Yes, they were. In this study, the pseudo ATLID L1 data were created by J-simulator. In that calculation, multiple scattering is taken into account by using the η-factor.

Line 311 : "..highliting.." ==> "..highlighting.."

=> We have corrected it as indicated.

**Conclusion**

Line 412 : "...will be released as JAXA's L2 ATLID standard products."

=> We have corrected it as indicated.

**References**

Check the references to the other special issue papers and update them if appropriate.

 =>We reviewed the references and made corrections.

---

## Author Comment (AC2)

**Reviewer 2**

The manuscript describes the JAXA algorithm to derive lidar L1 and L2 data from EarthCARE's Atmospheric Lidar (ATLID)a The paper provides an important contribution with respect to JAXA's EarthCARE data and should be published after addressing some mainly minor or editorial points.

**Abstract**

L15: '… were estimated by our developed optimization method…' should it be rather be '… calculated'

=> Thank you for your comment. We modified it as follow,

"Aerosol optical properties such as extinction coefficient, backscatter coefficient, depolarization ratio, and lidar ratio at 355 nm were optimized to ATLID L1 data by the method of maximum likelihood."

L27f: '37% of misclassification for dust' – should not be addressed in the Abstract,

=> We have removed it from the abstract as indicated.

but what causes this misclassification and why is it so high for dust?

=> Please see our response to your comments on LL372-374.

**Introduction**

The introduction has no clear thread but appears to me to have several jumps in topics. This is often confusing. The authors should consider rephrasing the introduction to achieve a more clear structure and argumentation.

=> Thank you very much for the comments. We have reviewed the part regarding the connection of sentences and added more explanations regarding the description to clarify the content. For example, we have added the following statements,

"Zhang et al. (2022) analyzed the simulation results of the Coupled Model Intercomparison Project Phase 6 (CMIP6) global models and indicated that 20% of radiative effect uncertainty for anthropogenic aerosols was associated with aerosol vertical distribution under clear-sky condition."

Some examples:

In large part it is written as it was dealing with satellite lidars ATLID and CALIPSO, but in ll59-61 retrievals are mentioned to derive microphysical and optical properties from wavelength combinations that are neither used by CALIPSO nor by ATLID. This is a bit confusing.

=> We modified it as follows,

"Multichannel lidar observations using a combination of polarization measurements, Rayleigh/Mie separation measurements, and/or various wavelength measurements, including those of ATLID, CALIOP, and ground-based lidar measurements, facilitate simultaneous understanding of various optical and microphysical properties of atmospheric particles."

L61: 'Furthermore, the extinction coefficients…' – further to what. Are the extinction coefficients not part of the optical properties? The authors should give some more information here.

=> We accidentally used an improper conjunction, which I removed.

L77: Based on the above background – Not clear to me what is meant with above background.

=> We removed it and made the description specific as follows,

"To determine the globally vertical distribution of optical properties of aerosols and clouds needed to assess

their climatic effects, we have developed atmospheric particle retrieval algorithms to generate JAXA L2 products using ATLID L1 data."

**Algorithm flow and products**

Figure 1: Are all the variables and abbreviations explained in the text? Please check.

=> We confirmed that all the variables and abbreviations are explained in the text.

L106: '…POP is of aerosol or cloud origin is …' – tow times 'is'

=> We have corrected it.

**Layer identification (Feature mask)**

This paragraph is hard to read and understand as some of the parameters are not introduced or their meaning is not introduced. The authors should consider to rephrase this paragraph in the sake for the reader to better understand. They should check carefully that all parameters are described.

=> Thank you very much for the comments. We have made significant revisions, such as being more specific about parameters that were ambiguous or not well explained, and changing phrases and paragraphs to improve readability.

L139 and following: What is meant with 'significant'?

=>We have modified it to be more descriptive as follows,

"(a) If the SNRs of $\beta_{atn,M}$ and $\beta_{atn,R}$ are lower than the threshold ($SNR_{th}$) or data is missing, no diagnosis is made in that layer (classified as "Invalid").

(b) If the SNR of $\beta_{atn,R}$ is greater than the $SNR_{th}$ and the SNR of $\beta_{atn,M}$ is lower than $SNR_{th}$, the layer is classified as "Clear-sky."

(c) The SNR of $\beta_{atn,M}$ is greather than the $SNR_{th}$, the layer is classified as "Aerosol," "Cloud," or "Surface. In this study, the value of $SNR_{th}$ is set to 3."

LL158-160: Not clear to me. What is Pp?

=> We deleted Pp, PR, PM and modified it to be more specific as follows,

"If the SNR of $\beta_{atn,R}$ of the target layer is greater than $SNR_{th}$, the Equation 3(b) is used as the criteria; otherwise, Equation 3(a), is used. The advantage of using $\beta_P$ as a diagnostic parameter is that it can eliminate attenuation due to clouds/aerosols from the ATLID to the target layer. However, the calculation of the $\beta_P$ requires $\beta_{atn,R}$ in addition to $\beta_{atn,M}$, which has a disadvantage in that it cannot be diagnosed if the SNR of $\beta_{atn,R}$ is insufficient."

L168: 'aeroso.l' typo

=> This is a typo. This appears to have occurred during conversion from word to PDF, I will contact AMT.

**Aerosol type classification**

This paragraph is not fully clear to me. E.g. the required optical properties and size distributions from AERONET measurements. Are they used as a general input for different aerosols? Or is the closest measurement in space and time required for the algorithm? The authors should think of rephrasing part of this paragraph.

=> Thank you very much for the comments. We changed the sentences to the following sentences.

"The cluster, which is fine-mode dominated and light-absorbing aerosol, derived from the cluster analysis of the observations of AERONET sites located from 25°S to 35°S and 0°E to 40°E, which cover the source

regions of African biomass burning aerosols, is assumed as the smoke type. The cluster, which is fine-mode dominated and light-absorbing aerosol, derived from the cluster analysis of the observations of AERONET sites located from 20°N to 40°N and 100°E to 125°E, which cover the source regions of Asian air pollution, is assumed as the pollution type. The marine and pristine types are derived from the cluster analysis of the observations of AERONET island sites far from aerosol source regions, where the marine type has the largest particle size with single scattering albedo (SSA) of 0.98 and the pristine type has a smaller particle size than the marine type, with the SSA of 0.98."

L232: For sure these are the main aerosol types, but what about e.g. volcanic aerosols? Can the authors give a short assessment about the uncertainties in retrieving the radiative (and climate) impact, as well as the impact on Aerosol-Cloud-Interaction, if volcanic aerosols are classified as dust.
=> Thank you for your comment. We added the following sentences.
"The ATM consists of the major tropospheric aerosol types and the optical properties of the ATM are used in the radiative transfer calculation to estimate aerosol radiative effect (Yamauchi et al., 2024). For example, volcanic ash and stratospheric aerosols, which are not included in the ATM, are misclassified as one of the aerosol types of the ATM. This misclassification is one of the uncertainties of the estimated aerosol radiative effect. However, this uncertainty is smaller than the other uncertainties in the estimation of aerosol radiative effect, because the proportions of the volcanic ash and stratospheric aerosols are considerably small in the total aerosol amounts."

L243: Is the spheroid model sufficient for dust?
=> It has been reported that the spheroidal model does not fully reproduce dust lidar observations (e.g., Müller et al., 2013), so that δa and Sa of dust type are referred to as the averaged values of the Raman lidar observations in this study.

Müller, D., Veselovskii, I., Kolgotin, A., Tesche, M., Ansmann, A., and Dubovik, O.: Vertical profiles of pure dust (SAMUM-1) and mixed smoke-dust plumes (SAMUM-2) inferred from inversion of multiwavelength Raman/polarization lidar data and comparison to AERONET retrievals and in-situ observations, Appl. Opt., 52, 3178–3202, https://doi.org/10.1364/AO.52.003178, 2013.

LL157-159: There are also studies about long-range transported dust and on Arabian dust. The authors should consider to discuss them as well.
=>We added the following sentence.
"δa of 0.24 ± 0.02 and Sa of 47 ± 8 sr for transported dust (Haarig et al., 2022) are also within the range of the dust type of the ATM."

Haarig, M., Ansmann, A., Engelmann, R., Baars, H., Toledano, C., Torres, B., Althausen, D., Radenz, M., and Wandinger, U.: First triple-wavelength lidar observations of depolarization and extinction-to-backscatter ratios of Saharan dust, Atmos. Chem. Phys., 22, 355–369, https://doi.org/10.5194/acp-22-355-2022, 2022.

**Planetary boundary layer height**
The first paragraph reads more like a general introduction to this topic. The authors should consider to move this to the introduction to motivate the need to develop the corresponding algorithms.
=> Thank you for your comment. We moved it to the introduction.

L275: What is mean by 'results in a small'?

=> To clarify this sentence, the following modifications have been made, including around the pertinent text. "Due to relatively larger Rayleigh scattering at 355 nm (approximately five times larger than that at 532 nm), the proportion of Mie scattering by aerosols in the PBL in lidar backscatter becomes relatively smaller. In other words, the difference in 355 nm backscatter signals for aerosol-rich PBL and aerosol-poor free troposphere (FT) can be smaller. The larger Rayleigh scattering at 355nm also produces greater signal attenuation, resulting in the lower signal difference between the PBL and the FT. It should be noted that the signal attenuation due to Rayleigh scattering near the top of the PBL is generally larger for spaceborne lidar observations than for ground-based lidar observations. Thus, the detection of PBLH by spaceborne lidar observation at 355nm is more challenging than in the past, even when the WCT method is applied (Kim et al., 2021). "

**Results and Discussion**

Figure 5: Why do the cases for 0.3 km and 1 km horizontal resolution data look so different? Have different cases been used? Or rather, it looks like the example for 0.3 km is just the excerpt for the part form 0 to about 1.31 in the 1 km data. If this is the case it is inconsistent with Figure 3.

=> Thank you for pointing out that we had pasted a figure for a different case for 0.3 km. We have corrected it to the correct figure. The correct figure is as follow.

[Figure]

Figure 5. Feature mask products estimated for data with/without signal noise for cloud predominant scenes (Figure 3) and its true or false values. The upper/lower figures show the results for the 0.3/1 km horizontal resolution data.

LL361-368: The authors should consider to move that to the retrieval part. – Is multiple scattering considered in their algorithm?

=> We moved this paragraph to the end of section 3.2. The algorithm already implements the method using the η-factor, but since we are dealing with aerosols, we performed various estimations assuming η = 1 (no multiple scattering).

L372: What is meant by 'layers' here?
=>We have modified it as "The total number of layers classified as Dust type".

LL372-374: What is the main reason for the misidentification?
Figure 9: It is interesting to see, that dust is quite well detected (although many false classification parts), but pristine is not really captured well. It would be interesting to see; which aerosols have a high certainty to be classified and which tend to be more often misclassified. And what are the aerosols they are mainly classified at.
General remark: It would have been interesting to see also a case with different aerosol types just only one example with dust as the dominant aerosol type.
=> We are very pleased with your interest in the results of this analysis. In this algorithm, aerosol type is determined by searching a table for each modeled aerosol type (Figure 2) against the retrieved lidar ratio and depolarization ratio values. Thus, aerosol type misclassification is directly reflected by the errors in the lidar ratio and depolarization ratio. As shown in section 3.4, the error in the lidar ratio is considerably larger than that in the depolarization ratio, about 25 sr in RMSE. Therefore, the retrieved lidar ratio for the dust layer deviates from the lidar ratio-depolarization ratio area for "dust" in Fig. 2, resulting in cases of misclassification as dusty-mixture or unknown. Similarly, pristine is misclassified as pollution or marine due to the errors in the lidar ratio. Simply, for this algorithm, the classification of aerosol type with a wider lidar-ratio - depolarization ratio area (Fig. 2) is more robust against errors in lidar ratio and depolarization ratio. We have added this discussion to section 4 and conclusion.

LL383-385: The authors already mentioned that that the cloud algorithm is presented in separate publications. To include it in four lines in this manuscript does not give a meaningful description of the cloud algorithm nor a meaningful contribution to this manuscript. The authors should consider to remove it.
=> As you commented, the cloud retrieval is discussed in detail in a separate paper, but we think it would be beneficial to the reader to provide some specific information. On the other hand, it was not the appropriate section to describe, and we have moved it to the end of section 3.3, which describes the methodology.

Conclusion
L401: It has to be made clear that this is not real ATLID L1 data but simulations. Please also check throughout the manuscript.
=> We modified the sentence as follows,
"The performance of the algorithms was demonstrated using pseudo-ATLID L1 data with realistic signal noise, which were calculated according to the ATLID specification for the aerosol or cloud dominated cases by Joint-Simulator."

Conclusion and Abstract are very similar. The authors should think about rewriting the one or the other. Instead of repeating it would have been interesting to give a discussion on the main contributors to the uncertainties.
=> Both Abstract and Conclusion have been revised. A description of error factor was added to the conclusion, and a description of the EarthCARE mission was added to the abstract. We also reduced the duplication of information between Abstract and Conclusion. The revised "Abstract" and "Conclusion" are as follows.

Revised Abstract
The Earth Cloud Aerosol and Radiation Explorer (EarthCARE) is a joint Japanese-European satellite

observation mission for understanding the interaction between cloud, aerosol, and radiation processes and improving the accuracy of climate change predictions. The EarthCARE satellite was equipped with four sensors, a 355 nm high-spectral-resolution lidar with depolarization measurement capability (ATLID) as well as a cloud profiling radar, a multi-spectral imager, and a broadband radiometer, to observe the global distribution of clouds, aerosols, and radiation. In this study, we have developed algorithms to produce ATLID Level 2 aerosol products using ATLID Level 1 data. The algorithms estimated the following four products: (1) Layer identifiers such as aerosols, clouds, clear-skies, or surfaces were estimated by the combined use of vertically variable criteria and spatial continuity methods developed for the CALIOP (Cloud-Aerosol Lidar and Infrared Pathfinder Satellite Observation) analysis. (2) Aerosol optical properties such as extinction coefficient, backscatter coefficient, depolarization ratio, and lidar ratio at 355 nm were optimized to ATLID L1 data by the method of maximum likelihood. (3) Six aerosol types, namely smoke, pollution, marine, pristine, dusty-mixture, and dust were identified based on a two-dimensional diagram of the lidar ratio and depolarization ratio at 355 nm developed by cluster-analysis using the AERONET (AErosol RObotic NETwork) dataset with ground-based lidar data. (4) The planetary boundary layer height was determined using the improved wavelet covariance transform method for the ATLID analysis. The performance of various algorithms was evaluated using pseudo ATLID Level 1 data generated by Joint-Simulator (Joint Simulator for Satellite Sensors), which incorporates aerosol and cloud distributions simulated by numerical models. Results from applying the algorithms to the pseudo ATLID Level 1 data with realistic signal noise added for aerosol or cloud predominant cases revealed: (1) misidentification of aerosol and cloud layers was relatively low, approximately 10%; (2) the retrieval errors of aerosol optical properties were $0.08 \times 10^{-7} \pm 1.12 \times 10^{-7}$ m$^{-1}$sr$^{-1}$ ($2 \pm 34\%$ in relative error) for backscatter coefficient and $0.01 \pm 0.07$ ($4 \pm 27\%$ in relative error) for depolarization ratio; (3) aerosol type classification was generally performed well. These results indicate that the algorithm's capability to provide valuable insights into the global distribution of aerosols and clouds, facilitating assessments of their climate impact through atmospheric radiation processes.

Revised Conclusion

We developed algorithms to produce JAXA ATLID L2 products for aerosols and clouds using ATLID L1 data. The algorithms retrieve the following products: (1) feature mask; (2) particle optical properties such as extinction coefficient, backscatter coefficient, depolarization ratio, and lidar ratio at 355 nm; (3) target mask; and (4) planetary boundary layer height. The performance of the algorithms was demonstrated using pseudo-ATLID L1 data with realistic signal noise, which were calculated according to the ATLID specification for the aerosol or cloud dominated cases by Joint-Simulator.

The main findings of this simulation study are as follows: (1) the misidentification of the aerosol and cloud layers by the feature mask algorithm was relatively low (approximately 10%). (2) The retrieval errors of the AOP were $0.08 \times 10^{-7} \pm 1.12 \times 10^{-7}$ m$^{-1}$sr$^{-1}$ ($2 \pm 34\%$ relative error) for the backscatter coefficient and $0.01 \pm 0.07$ ($4 \pm 27\%$ relative error) for the depolarization ratio of aerosols; the relative errors of the extinction coefficient and lidar ratio were worse than those of the backscatter coefficient and depolarization ratio. (3) The aerosol-type classification generally performed well, with 37% misclassifications for the dust type, although there were more misclassifications for the pristine type. These misidentifications were mainly due to errors in the estimation of the lidar ratios. Accurate estimation of the lidar ratio as well as depolarization ratio is essential for aerosol type classification. (4) PBLH retrieval using the WCT method with ATLID L1 data was feasible. These results indicate that the algorithm's capability to provide valuable insights into the global distribution of aerosols and clouds, facilitating assessments of their climate impact through atmospheric radiation processes.

The validation of these aerosol and cloud products is essential, and JAXA and ESA will conduct validation

using various platforms, such as airborne, shipborne, and ground-based measurements, and various observation instruments, including lidars. The validation studies include assuarance of the product accuracy and the improvement of the L2 algorithms mentioned above, including the verification and improvement of various assumptions (e.g., optical models of aerosols and clouds) and the criteria used in the algorithms.

To understand the climatic effects of aerosols, it is essential to understand the properties of individual aerosol components as well as the properties of total aerosols such as the JAXA ATLID L2 products described above. Therefore, we are currently developing an algorithm to estimate the extinction coefficients of dust, sea salt, carbonaceous (light-absorbing particles), and water-soluble aerosols at 355nm using the difference in the depolarization and light absorption properties of each aerosol component from the L2 products and ATLID L1 data. We are also currently developing an ATLID-MSI synergy algorithm to retrieve the vertical mean mode-radii of dust and fine-mode aerosols, and the extinction coefficients of the abovementioned four aerosol components. These algorithms are being developed based on the aerosol component retrieval algorithms that have been developed for the analysis of CALIOP and ground-based lidar data. (Kudo et al. 2023; Nishizawa et al. 2007, 2008, 2011, 2017). The aerosol component products derived by these algorithms using ATLID and MSI data are planned to be released as JAXA's L2 products and are expected to enhance the scientific value of the EarthCARE mission.

L413: What are the standard algorithms?
=> JAXA divides L2 products into two categories ("Standard product" and "Research product"), and the algorithm that generates the standard product is called as "Standard algorithm". Since these categories are not so important in this paper and confusing to readers, we have decided to remove the descriptions of these categories.

L413f: 'we have been developing an algorithm to estimate…' – Was that shown in this manuscript? If it is of importance to mention it here, it should be better introduced and motivated.
=> Thank you for your comment, we have added motivation and modified the description as follows,
"To understand the climatic effects of aerosols, it is essential to understand the properties of individual aerosol components as well as the properties of total aerosols such as the JAXA ATLID L2 products described above. Therefore, we are currently developing an algorithm to estimate the extinction coefficients of dust, sea salt, carbonaceous (light-absorbing particles), and water-soluble aerosols at 355nm using the difference in the depolarization and light absorption properties of each aerosol component from the L2 products and ATLID L1 data. We are also currently developing an ATLID-MSI synergy algorithm to retrieve the vertical mean mode-radii of dust and fine-mode aerosols, and the extinction coefficients of the abovementioned four aerosol components. These algorithms are being developed based on the aerosol component retrieval algorithms that have been developed for the analysis of CALIOP and ground-based lidar data. (Kudo et al. 2023; Nishizawa et al. 2007, 2008, 2011, 2017). The aerosol component products derived by these algorithms using ATLID and MSI data are planned to be released as JAXA's L2 products and are expected to enhance the scientific value of the EarthCARE mission."

---

## Referee Report (RR1)

Review of "Algorithm to retrieve aerosol optical properties using lidar measurements on board the EarthCARE satellite", Revision #1.

By T. Nishizawa et al.

General Remarks
===============

This paper provides a useful concise overview of the JAXA EarthCARE lidar products and will be a useful reference for the community. I recommend publication. There are, however, several mainly editorial issues that need to be addressed.

The authors have done a good job responding to the issues I raided in my earlier review. I have only noted a couple of editorial issues.

1) Line 86: "..are estimated from the ESA" ==> "..are estimated by ESA.."

2) Line 210: "..but the SNR is not sufficiently small". "...but the SNR is not sufficiently large".

---

## Author Response (AR2)

**Reviewer 1**

This paper provides a useful concise overview of the JAXA EarthCARE lidar products and will be a useful reference for the community. I recommend publication. There are, however, several mainly editorial issues that need to be addressed. The authors have done a good job responding to the issues I raided in my earlier review. I have only noted a couple of editorial issues.

1) Line 86: "..are estimated from the ESA" ==> "..are estimated by ESA.."

2) Line 210: "..but the SNR is not sufficiently small". "...but the SNR is not sufficiently large".

=> Thank you for pointing out. We have made the correction as noted.

**Reviewer 2**

=> No requests on revision were made (no referee report was found).

Along with revisions addressing the reviewer's comments above, we have made the following technical corrections. These corrections do not directly relate to the content of the paper.

Line 5-7: The affiliation description has been corrected.

Line 457-459: A description regarding data availability has been added. Accordingly, one reference (Roh et al. 2023b) has been added.

Line 482-485: We found missing programs regarding financial support and added them as follows,
"This research was supported by the EarthCARE satellite study commissioned by the Japan Aerospace Exploration Agency (grant no. 23RT000223), a Grant-in-Aid for Scientific Research (KAKENHI) (project numbers: JP18KK0289, JP17H06139, JPS17H04477, JP15H01728, JP15H02808, JP22K03721, JP24H00275, and JP25220101), and Collaborative Research Program of the Research Institute for Applied Mechanics, Kyushu University (Fukuoka Japan)."

Line 1148-1149: We found a typo and corrected it as follows.
"Oikawa E., Nakajima T., Inoue T., Winker D.: A study of the shortwave direct aerosol forcing using ESSP/CALIPSO observation and GCM simulation, J. Geophys. Res. Atmos., 118, 3687-3708, https://doi.org/10.1002/jgrd.50227, 2013."